# The Confidence of Undergraduate Dental Students When Undertaking Indirect Restorations

James Puryer * , Shaikho Khalaf and Maryam Ilkhani

Bristol Dental School, University of Bristol, Lower Maudlin Street, Bristol BS1 2LY, UK

* Correspondence: james.puryer@bristol.ac.uk; Tel.: +44-0117-342-4425

**Abstract:** The General Dental Council expects United Kingdom graduating dentists to be competent at providing crown and bridge treatments and graduating students should feel confident to undertake these treatments. The aim of this study was to explore the confidence of undergraduate dental students when providing crown and fixed dental prosthesis treatments. An anonymous questionnaire was distributed to all ($n$ = 198) undergraduate students in Years 3 to 5 at Bristol Dental School. The questionnaire comprised confidence interval ranked responses, and respondents' data was analysed using SPSS. The response rate was 76%. Undergraduate perception of confidence increased throughout the three years of study ($p < 0.01$). There was a strong association between the amount of exposure students had to certain treatment stages and their perceived confidence levels. Confidence levels may be increased further with increased clinical experience.

**Keywords:** undergraduate; indirect; restoration; crown; prosthesis; confidence

## 1. Introduction

### 1.1. What Are Indirect Restorations

Indirect restorations are those fabricated outside of the mouth by a dental technician in a laboratory using dental impressions of the prepared tooth, and include crowns, bridges, inlays and veneers. They may also be fabricated using computer-aided design and manufacturing (CAD/CAM) technology. Indirect restorations are commonly placed on heavily restored teeth, worn teeth and posterior teeth following endodontic treatment [1].

Indirect restorations have been shown to be an excellent choice of restoration in posterior Class I and II cavities due to their high success rate (91% in composite and 96% in ceramic) [2], and they are a popular choice of restoration among practitioners in the United Kingdom (UK) [3]. Furthermore, there may be a shift in the way individuals with missing teeth are perceived, with an increased importance placed on a more aesthetic dental appearance. The stigma attached to missing teeth motivates patients to seek restorations that imitate natural tooth structure [4].

The success of an indirect restoration relies mostly on a clinician's ability to perform the treatment to a high standard, based on underlying science [5]. Additionally, it is important that clinicians can communicate the restoration requirements to the laboratory using clear instructions [6].

### 1.2. Why Must Dental Undergraduates Learn to Place Indirect Restorations

All United Kingdom (UK) dental graduates are required to meet the learning outcomes identified in the 2015 General Dental Council (GDC) document "Preparing for Practice" [7]. Learning outcome 1.14.4 states that qualifying dentists should be able to, "Where appropriate, restore the dentition using the principle of minimal intervention, to a standard that promotes the longevity of the restoration or prostheses". This is followed by learning outcome 1.14.5, which states that qualifying dentists should

be able to "Manage restorative procedures that preserve tooth structure, replace missing or defective tooth structure, maintain function, are aesthetic and long lasting, and promote soft and hard tissue health". Furthermore, it is essential that the evidence-based indirect restoration teaching provided is sufficient to produce clinically excellent and thus confident "fully qualified beginners". This will help to maintain patient safety and the provision of high-quality care [8]. Although the GDC only governs dentistry within the UK, all dental education administrations within the European Union (EU) must reach similar standards to the GDC. Furthermore, a new framework has been developed that provides clear guidance and support for dental education and training on a European scale, incorporating explicit learning outcomes [9]. European graduates should be prepared to provide oral health care for patients aspiring to high standards of professional and clinical care.

*1.3. What Is Meant by Confidence*

Confidence in providing clinical care for patients is considered an important education outcome for dental undergraduates [10]. Confidence can be defined as "a state of certainty in the success of a particular behavioural act", and influences whether an individual is willing or not to undertake a specific activity [11]. Competence, which represents an individual's ability and is based on that individual's previous experience of the task, does not always directly relate to confidence [12]. Although students may have the necessary skills and context, their internal perception of their ability may not give them confidence to carry out the procedure. There are contrasting views as to whether having high levels of confidence can correlate with successful performance [13,14]. There is a risk that overconfident students may put patients at risk by attempting procedures beyond their skill level.

Although increasing a student's clinical experience of indirect restorations will not necessarily mean that a student will become competent, it is essential that students receive sufficient clinical exposure to such treatments; else they are unlikely to develop either competence or confidence [15]. Irrespective of the amount of clinical experience gained whilst a dental undergraduate, it is essential that students develop insight and an accurate self-assessment of their own competence levels and associated confidence. It must be remembered that perceived confidence is individualistic and may be temporary in nature, with both feedback from staff and clinical outcome affecting levels of confidence. Further training and clinical experience can be sought after graduation where necessary. Upon graduation, dentists need the skills to be able to target their 'weak' areas through training by using portfolios, reflection and personal development plans [16].

A study of final year dental students at the Cardiff School of Dentistry found lower confidence levels associated with complex procedures that were least practiced [16]. These results are supported by a later study undertaken in Riyadh, Saudi Arabia [17]. The findings of these two studies were reflected in a study carried out in England, Northern Ireland and Wales where it was found that 51% of Foundation Trainers who supervise dentists during their first year post-graduation reported that the overall standard of those entering foundation training was 'unsatisfactory'. Standards in key clinical areas were considered unsatisfactory by large proportions of respondents with 85% considering standards unsatisfactory in 'crown and bridge' [18]. However, the findings of that study must be taken with caution as it received only a 28% response rate. Previous studies carried out at Bristol Dental School have looked at undergraduate confidence levels within endodontics and prosthodontics. The prosthodontics study found that perceived confidence of students increased as they progressed through the course [19]. The endodontic study also showed that the confidence of dental students was directly related to their progression throughout the course, with the majority highly confident at performing root canal treatments in a clinical setting towards their time of graduation [20]. Students' confidence levels also were directly related to their amount of clinical experience. In contrast to the majority of studies that found increased clinical experience was associated with higher confidence levels, one later UK study found a poor correlation between reported confidence and the number of extractions carried out post-graduation [21].

The relationship between increased confidence and further clinical experience has also been found amongst medical students and junior doctors [14,22].

Differences in teaching methods may influence confidence, with students who experience clinical demonstrations exhibiting higher levels of confidence as opposed to students who only received theoretical teaching [10]. Problem based learning (PBL) is a recent addition to teaching methods and since it was introduced to clinical dental teaching, research suggests that the methodology of having a student-centred, small grouped, PBL approach produced practitioners of a higher standard compared to those taught purely by traditional teaching methods [23]. Additionally, it has been shown that students who have experienced clinical demonstrations before their own supervised treatment of patients felt more confident than those students who only receive PBL teaching [10].

The University of Bristol Dental School follows a teaching strategy whereby the main objective is to facilitate student development in furthering their clinical learning, technical competence and professionalism. It uses an integrated approach, combining PBL in the form of library and critical appraisal project, and online teaching resources alongside a more formal education. PBL related to indirect restorations is delivered via online tutorials and quizzes that students can undertake at their own pace. The formal education related to indirect restorations (in addition to subsequent clinical chairside teaching) comprises:

- Year-2: 6 × 45-min lectures, 11 × 2 h 45 min practical sessions
- Year-4: 7 × 45-min lectures, 15 × 2 h 45 min practical sessions

In Year-2, the teaching of indirect restorations covers the preparation and fit of a full-gold crown, all-ceramic crown and a porcelain fused to metal crown. Students are shown a video demonstration of the tooth preparation needed, prior to undertaking the preparation themselves on dental manikins. It has been shown elsewhere that students produce more accurately tapered preparations when using real time video demonstrations [24]. The Year-4 teaching course covers the preparation of conventional fixed dental prostheses, resin-retained fixed dental prostheses, inlays, onlays, post-retained crowns and veneers. This course follows a similar format to that of Year-2 whereby, following a video demonstration, students undertake each procedure on a dental manikin. Following completion of each of those two courses, students are formally assessed to ensure competence as per the GDC guidelines, which state, "Students must provide patient care only when they have demonstrated adequate knowledge and skills. For clinical procedures, the student should be assessed as competent in the relevant skills at the levels required in the pre-clinical environments prior to treating patients" [8]. If successful in theses summative assessments, students may go on to perform these procedures on patients. Having progressed onto patient clinics, students receive continuous formative feedback following each clinical treatment session from their supervising clinicians. Currently, upon qualifying, students are expected to have completed a minimum of eight indirect restorations on their patients. It is hoped that students then have both the confidence and competence to undertake indirect restorations at a 'safe beginner' level [7].

The aim of this study was to explore the perceived confidence levels of undergraduate dental students in Years 3 to 5 at Bristol Dental School when performing indirect restorations.

## 2. Materials and Methods

Full ethical approval from the Faculty of Health Sciences Committee for Ethics was obtained prior to the study. (Approval number 60710).

An anonymous cross-sectional survey of all dental undergraduates ($n = 198$) studying in Years 3 to 5 at the University of Bristol was carried out. There were no exclusion criteria. A questionnaire was developed, based upon two previous studies [18,19], that utilized confidence interval responses using a Likert scale to explore students' views on confidence whilst undertaking various stages of indirect restorations, with a scores ranging from '1' = "Not at all confident" to '10' = "Extremely confident." Hard copy questionnaires were distributed in timetabled lectures between January and April 2018.

For consent purposes, potential participants were e-mailed prior to the lectures with a Participant Information Sheet (PIS) so that they had time to consider if they wished to participate. A consent form was not considered necessary as consent was implied by the participant choosing to take part in the study. Students were allowed to withdraw at any point whilst completing the questionnaire, and this was made clear in the PIS. Data from the quantitative questions was analyzed using Statistical Package for Social Sciences (SPSS).

## 3. Results

There was a 76% response rate with $n = 152$ students completing the questionnaire. This comprised $n = 42$ (63%) students from Year-3, $n = 52$ (81%) students from Year-4 and $n = 58$ (85%) students from Year-5.

Levels of overall reported confidence increased ($p < 0.001$) as students progressed through the course, with mean confidence levels for students being 3.7 (Year-3), 5.8 (Year-4) and 7.2 (Year-5). This overall increase in confidence levels as students progressed was also seen in almost all specific indirect restoration procedures (Table 1). However, there was no statistically significant difference found between the results for four specific procedures: Mixing zinc phosphate cement ($p = < 0.863$); mixing zinc polycarboxylate cement ($p = < 0.597$); mixing Zinc Oxide Eugenol cement ($p = < 0.820$); and mixing silicone for the creation of a putty stent ($p = < 0.135$).

**Table 1.** Mean undergraduate confidence levels (and standard deviation) for individual indirect restoration procedures where '1' = 'not at all confident' and '10' = 'extremely confident'.

| How confident do you feel? | Year-3 | Year-4 | Year-5 | *p*-Value |
|---|---|---|---|---|
| Making primary impressions for study models using alginate? | 6.45 (1.38) | 7.98 (1.65) | 8.72 (1.15) | 0.000 |
| Choosing the correct size of stock tray when taking an impression? | 7.69 (1.44) | 8.54 (1.38) | 8.94 (1.02) | 0.001 |
| Assessing the restorability of a patient's tooth? | 4.69 (1.24) | 5.98 (2.07) | 7.05 (1.34) | 0.000 |
| Determining the appropriate restoration to place on a tooth? | 5.93 (1.18) | 6.52 (1.91) | 7.12 (1.55) | 0.000 |
| Taking a master impression using light and heavy bodied silicone? | 4.90 (1.46) | 7.35 (2.20) | 7.79 (1.32) | 0.000 |
| Taking a bite registration using a silicone impression material? | 4.00 (1.21) | 8.17 (1.80) | 9.00 (1.18) | 0.000 |
| Taking a facebow registration? | 1.43 (1.04) | 4.46 (2.40) | 5.05 (2.10) | 0.000 |
| Choosing which luting cement to use? | 3.81 (1.61) | 6.35 (2.13) | 6.31 (2.20) | 0.000 |
| Taking a bite registration using modelling wax? | 6.02 (1.52) | 7.00 (2.70) | 8.22 (1.86) | 0.000 |
| Creating and fitting a temporary crown using Maxitemp? | 5.19 (1.86) | 7.12 (2.37) | 7.45 (1.90) | 0.000 |
| Creating and fitting a preformed metal temporary crown? | 6.19 (1.35) | 5.06 (2.73) | 6.74 (1.73) | 0.004 |
| Mixing Aquacem? | 7.69 (1.66) | 8.52 (1.18) | 8.71 (1.50) | 0.046 |
| Mixing Panavia? | 5.19 (2.46) | 6.73 (2.73) | 8.00 (1.92) | 0.000 |
| Mixing Zinc Phosphate cement? | 7.02 (1.89) | 7.25 (2.72) | 6.91 (2.18) | 0.863 |
| Mixing Zinc Polycarboxylate cement? | 7.76 (1.72) | 7.50 (1.90) | 6.98 (2.18) | 0.597 |
| Mixing Zinc Oxide Eugenol cement? | 7.38 (1.90) | 7.50 (1.87) | 7.43 (2.06) | 0.820 |
| Mixing silicone for the creation of a putty stent? | 8.12 (2.62) | 8.10 (1.77) | 8.72 (1.70) | 0.135 |
| Preparing a tooth for a conventional fixed dental prosthesis? | 1.19 (0.94) | 5.40 (2.61) | 6.60 (2.26) | 0.000 |
| Preparing a tooth for a resin retained fixed dental prosthesis? | 1.00 (0.00) | 7.27 (1.95) | 8.14 (1.73) | 0.000 |
| Preparing a tooth for a full gold crown? | 6.12 (1.60) | 7.08 (2.08) | 8.10 (1.33) | 0.000 |
| Preparing a tooth for a PFM crown? | 5.07 (1.93) | 6.79 (1.92) | 7.64 (1.66) | 0.000 |
| Preparing a tooth for a ceramic veneer? | 1.10 (0.62) | 5.96 (2.36) | 6.38 (1.92) | 0.000 |
| Preparing a tooth for a Porcelain Jacket Crown? | 2.10 (1.65) | 6.27 (2.27) | 6.64 (1.93) | 0.000 |
| Preparing a tooth for a gold onlay or inlay? | 1.14 (0.65) | 5.63 (2.62) | 7.38 (1.77) | 0.000 |
| Preparing a tooth for a composite onlay or inlay? | 1.38 (1.53) | 6.38 (2.23) | 7.12 (1.83) | 0.000 |
| Preparing a tooth for a ceramic onlay or inlay? | 1.55 (1.50) | 6.00 (2.43) | 7.09 (1.92) | 0.000 |
| Selecting the correct shade for the restoration? | 8.21 (2.45) | 8.25 (1.47) | 8.07 (1.45) | 0.021 |
| Assessing the fit of an indirect restoration? | 4.86 (2.25) | 7.38 (1.99) | 8.22 (1.23) | 0.000 |
| Using articulating paper to assess the occlusion? | 7.40 (2.05) | 7.94 (1.60) | 8.62 (1.58) | 0.001 |
| Using shimstock to assess the occlusion? | 1.62 (1.48) | 4.87 (2.98) | 6.53 (3.03) | 0.000 |
| Adjusting an indirect restoration chairside? | 2.12 (1.82) | 5.88 (2.67) | 7.36 (1.77) | 0.000 |
| Polishing an indirect restoration chairside? | 2.69 (2.56) | 6.33 (2.44) | 7.28 (1.93) | 0.000 |
| Cementing an indirect restoration? | 2.90 (2.45) | 7.13 (1.91) | 8.29 (1.55) | 0.000 |

## 4. Discussion

This study found that overall levels of reported student confidence increased with course progression, which supports the findings of previous UK studies [16,19,20]. When looking at specific indirect procedures (Table 1), it was again found that there was an overall trend for increasing confidence as students progressed, although it was found that final-year students felt more confident at undertaking some procedures more than others. The procedures in which final-year students felt most confident were taking a bite registration with silicone bite registration material, choosing the correct stock tray and making primary alginate impressions for study models. This is unsurprising, as these procedures would most likely have been carried out many times previously, with students gaining good clinical experience with them. Students started undertaking these procedures in Year-2 and there is much overlap in this teaching with other parts of the curriculum, such as choosing the correct sized stock tray in removable prosthodontics teaching. It is also unsurprising that students in Year-3 did not show high levels of perceived confidence to carry out a number of defined procedures (for example, preparing a tooth to receive a conventional fixed dental prosthesis), as they would not yet have received formal teaching on these procedures.

It is reassuring that final year students had high confidence levels in more complex procedures such as the preparation of a full gold crown (8.1) and a porcelain-fused-to-metal crown (7.64). It is also unsurprising that final-year students felt more confident preparing teeth for a resin-retained fixed dental prosthesis (8.14), than for a conventional fixed dental prosthesis (6.6), which is potentially clinically more challenging. The procedure that had the lowest level of reported confidence by final-year students was taking a facebow registration (5.05). This procedure is taught in Year-4 of the programme and involves a formal lecture, a video tutorial and a 'hands-on' practical session. Students may be reporting low levels of confidence with taking a facebow recording, as they may not have the opportunity to undertake the procedure in clinic subsequently, possibly due to it not being deemed clinically essential for all indirect restoration preparations by their clinical supervisors. As this is a 'non-invasive' procedure, there may be an argument for future students undertaking a facebow record for *all* indirect restorations or for practicing the procedure on fellow students to ensure they have sufficient clinical experience to become confident with this skill. The four specific procedures that did not follow the pattern of being associated with increased confidence levels with student progression were all related to mixing of materials with high confidence levels being expressed by Year-3 students. This could be attributed to the fact that these procedures are often carried out by Year-3 students when they assist Year-5 students, whilst carrying out fourhanded dentistry. Year-3 students may also feel confident with handling and mixing dental materials, as these procedures may be seen as having little clinical risk to a patient when compared to undertaking irreversible tooth preparations. Students in Year-5 reported overall lower confidence levels in taking a shade of a restoration than those in Year-3. This may be due to the teaching being delivered within Year-3 and indicates that there may be a need to revisit teaching of this important subject later in the curriculum. Students may also be apprehensive of the financial implications of selecting the 'wrong' shade and the restoration having to be remade.

The school has recently introduced several dental manikins that can be set up and used in clinic by students if they have no patient booked or if their patient cancels their appointment. Rather than waste this valuable clinical time, students are encouraged to undertake restorative procedures, including indirect restoration preparations on one of these dental manikins, following which they will receive feedback from supervising clinicians. The benefit of learning on dental manikins within a controlled environment is that it removes many of the pressures associated with managing a patient in the clinical setting. This allows students to invest all their energy into perfecting their preparations and mastering their manual dexterity, without having to worry about additional patient factors. Previous studies have shown that students think highly of simulated manikin training [25–27] and their use can help to further levels of student confidence.

*4.1. Study Limitations*

Overall, this study has met its aims and objectives, but it does have some limitations. The response rate of 76% means that there may be some selection bias in respondents and may not be representative of all students. However, it has been suggested that a response rate of 60% can still be satisfactory [28] for this type of study.

Students' perceptions of confidence levels are subjective, and there was possible individual variation in interpretation of the questionnaire as to where on the 1–10 scale the cut off was for being 'confident' and 'not confident'. In addition, confidence is also a complex construct and it may have some relationship to individual personality traits and other factors as well as a student's ability to accurately assess their competence [29].

The results of this study are specific to one UK dental school. Each dental school will have its own curriculum and methods of teaching, thus students will have differing clinical and educational experiences between schools, and so it would be incorrect to assume that these results can be generalized. It would also be wrong to generalise these results for all students as the reported levels of perceived confidence relate to the *mean levels* across year cohorts of students. Equally, if not more important, are the perceived levels of confidence of individual students. It is hoped that students with low levels of perceived confidence will have the insight and maturity to seek additional opportunities for further clinical experience, hopefully increasing confidence levels. Although clinical confidence (the self-perceived ability to deal with clinical scenarios and undertake clinical treatment) does not necessarily correlate with competency, it is a pre-requisite for students to be able to fully participate in clinical activity [22].

However, despite these limitations, this study provides a contemporary benchmark of the confidence levels of dental students when undertaking indirect restorations. Further studies could be carried out at other UK dental schools or internationally to compare the experiences of their students. It would also be interesting to undertake studies to see how perceived levels of student confidence related to actual clinical competency. Previous studies within medicine have found little correlation between perceived confidence and actual competence [14,30].

*4.2. Implications*

It has previously been reported that there is widespread variation in the number of indirect restorations that UK dental schools expect students to complete prior to qualification [15,31,32]. Some schools reported that they do not have a minimum number of fixed dental prostheses that their students have to complete because their program is competency-based. Whether schools have minimum requirements or not, it is essential that students are clearly informed of any requirements with sufficient notice in order for them to be achieved [32]. Bristol Dental School is currently implementing a new curriculum and a major component of this will be to move all practical restorative manikin teaching to earlier within the curriculum. This means that students will complete all manikin teaching by the end of Year-3, providing them with two years of subsequent clinical experience. Alongside this, the number of completed indirect restorations by students expected of students is increasing, and various additional strategies are being implemented to facilitate this. It is unknown how many clincial experinces of repeating a clinical procedure a student will need before they feel confident to undertake that procedure independently. This is likely to vary between individuals. However, it is anticipated that this increased clinical exposure to restorative dentistry, including indirect restorations, will lead to a further increase in student perceived levels of confidence [19,20,33].

**5. Conclusions**

This study found that student levels of perceived confidence increased as they progressed though the course and gained further clinical experience, which supports the findings of previous studies. It is encouraging that the majority of students approaching graduation felt confident in undertaking

indirect restorations within a clinical setting. When designing curricula, schools should aim to provide students with as much clinical experience as possible to ensure high levels of confidence.

**Author Contributions:** Conceptualization, J.P.; methodology, J.P.; data collection and analysis, S.K. and M.I.; writing—original draft preparation, S.K. and M.I.; writing—review and editing, J.P.

**Funding:** This research received no external funding.

**Conflicts of Interest:** The authors declare no conflict of interest.

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
