# Peer review of "The Confidence of Undergraduate Dental Students When Undertaking Indirect Restorations"

_2571-8800, doi:10.3390/j2030019_

Round 1

Reviewer 1 Report

This study is based in 1 school in the UK and so is based around the GDC requirements. In Europe the publication by Field et al in Eur J Dent Ed 2017 - The Graduating European Dentist: A new undergraduate curriculum framework should be incoporated.

The discussion on confidence is necessary as is the need to explain the variations seen between individuals and between male and female. It is correct to differentiate the difference between confidence and competence and perhaps more should be made of this in the discussion.I was disappointed that all refeences were based within dentistry when there is such a large research base in other healthcare areas especially medicine. I was disappointed to see the paper by Oxley et al (2017) quoted without some note about the usefulness of the results of this study after noting the very poor response rate (around 28%)

The manuscript reports the formal teaching that is pursued however does not outline where the PBL is within this didactic and practical teaching.

The results are presented largely in tabulated form and although the sd is given perhaps it would have been useful to see also the range of responses? As discussed earlier there is individual variation and although the averages look ok how are the less confident (struggling) students viewing their own confidence especially those running up to graduation.

The discussion and conclusions were superficial but in reading them it confirmed my view that competence comes from understanding of the priniciples and mastering the techniques but confidence comes from repeating the procedure (experience). The question really is how many "experiences" does a student need before feeling confident to carry out such procedures independently - and this is likely to vary from individual to individual.

Author Response

Thank you for taking the time to review our paper and suggest ways in which it could be improved.

We feel that we have been able to address each of your comments as detailed in the attached document.

We hope that you will look favourably on these changes and recommend that this paper now be published within J.

Reviewer 2 Report

See attached file.

Author Response

(The authors gave the same response as above.)

Round 2

Reviewer 2 Report

Thanks for addressing the concerns of the peer-reviewers so quickly.  I have two minor comments that the type setter can correct.  Line 443 "be" is misspelled, and line 594 "generalized" is misspelled.